# Genetic predisposition to elevated BMI and adult asthma phenotypes in a Japanese population

Yohei Yatagai[1]*, Hisayuki Oshima[1], Yu Abe[1], Haruna Kitazawa[1], Hironori Masuko[1], Takashi Naito[2], Takefumi Saito[3], Tomomitsu Hirota[4], Mayumi Tamari[4], Emiko Noguchi[5], Tohru Sakamoto[6], Nobuyuki Hizawa[1]

1 Department of Pulmonary Medicine, Institute of Medicine, University of Tsukuba, Ibaraki, Japan, 2 Tsukuba Medical Center, Ibaraki, Japan, 3 National Hospital Organization Ibaraki Higashi National Hospital, Ibaraki, Japan, 4 Division of Molecular Genetics, The Jikei University School of Medicine, Research Center for Medical Science, 5 Department of Medical Genetics, Institute of Medicine, University of Tsukuba, Ibaraki, Japan, 6 Tsukuba Memorial Hospital, Tsukuba, Japan

* yyatagai@md.tsukuba.ac.jp

## Abstract

Obesity is a well-established risk factor for asthma, with genetic factors influencing both conditions. This study investigates the impact of genetic predisposition to increased body mass index (BMI) on adult asthma phenotypes. We recruited 1532 non-asthmatic healthy individuals and 779 adult asthma patients to assess the relationship between BMI-related genetic risk scores (BMI-GRS) and asthma. Among the 85 single nucleotide polymorphisms (SNPs) previously associated with BMI in Japanese populations, significant associations with BMI were confirmed for 6 SNPs in the healthy individuals. Using these, BMI-GRS was calculated for both groups. While asthma patients had higher BMI than healthy individuals (p=0.004), no significant difference in BMI-GRS was observed between the groups (p=0.56). A cluster analysis identified six distinct phenotypes of adult asthma patients: two overweight/obese clusters (one with elevated BMI-GRS, one without) and four non-obese clusters (with one showing significantly elevated BMI-GRS). This study demonstrates a genetic heterogeneity in the phenotype of adult asthma among a Japanese population, showing that genetic variants associated with BMI contribute to specific subtypes of asthma. Prospective longitudinal studies are essential to delineate the interactions between genetic predisposition, elevated BMI, subsequent changes in adiposity, and the evolution of asthma phenotypes, which would facilitate the development of mechanism-based therapeutic strategies tailored to genetically-defined patient subgroups.

## Introduction

It has long been known that there is a significant association between asthma and obesity [1]. Obesity and asthma represent major global health challenges, affecting over 1 billion and 262 million people worldwide, respectively [2,3]. In Japan, 33.0%

**Data availability statement:** All relevant data are within the paper and its Supporting Information files.

**Funding:** The author(s) received no specific funding for this work.

**Competing interests:** NO authors have competing interests.

of men and 22.3% of women exceed the Asian clinical threshold for overweight/obesity (BMI ≥ 25 kg/m²), while adult asthma prevalence is approximately 8–10% [4,5]. The intersection of these conditions is of particular clinical importance, as overweight and obese individuals have 1.17–1.62 and 1.92 higher odds of developing asthma, respectively, compared to those with normal BMI [6]. Obesity increases the risk of asthma through multiple well-characterized pathways [1,7]. Mechanical effects include reduced chest wall compliance and diaphragmatic excursion, leading to diminished lung volumes, airway dysanapsis, and hyperresponsiveness [8,9]. Systemic inflammation plays a central role, as adipose tissue secretes pro-inflammatory cytokines (TNF-α, IL-6) and altered adipokines (elevated leptin, reduced adiponectin) that promote Th1/Th17 responses and neutrophilic airway inflammation [7,10]. Metabolic pathways contribute through elevated asymmetric dimethylarginine (ADMA), which reduces nitric oxide bioavailability and impairs bronchodilation [11]. Hormonal influences, particularly dysregulated sex hormones in obese women, modulate airway inflammation and responsiveness [12]. Additionally, the gut-lung axis operates through obesity-related dysbiosis, altering microbial metabolites and increasing endotoxemia [1]. These mechanisms give rise to distinct obesity-associated asthma phenotypes, including late-onset, non-atopic, severe asthma in obese adults and early-onset allergic asthma that worsens with subsequent weight gain [13,14].

In Japanese populations, a recent genome-wide study linked BMI-related genes to asthma, [15] suggesting that increased BMI significantly contributes to the development of asthma, reinforcing its role as an asthma risk factor. Despite this established link between obesity and asthma, the influence of genetic predisposition to higher BMI on specific asthma phenotypes remains unclear.

In this study, we aimed to investigate the role of genetic predisposition to increased BMI on adult asthma phenotypes. By calculating a BMI-genetic risk score (BMI-GRS) based on SNPs associated with BMI, we attempted to clarify whether a genetic inclination toward higher BMI is associated with specific asthma phenotypes. Additionally, we explored whether genetic factors might distinguish certain subgroups of overweight or obese asthma patients. Understanding these relationships could provide valuable insights into the personalized management of asthma, particularly in overweight and obese populations.

## Materials and methods

### Study participants

The study participants were recruited from Tsukuba Cohort 1 and Tsukuba Cohort 2. Tsukuba Cohort 1 comprised 565 healthy individuals and 537 asthmatic patients, while Tsukuba Cohort 2 consisted of 967 healthy individuals and 242 asthmatic patients [16]. The definition of a clinical diagnosis of asthma was described in our previous study [17]. Asthmatic patients from both cohorts were enrolled from the University of Tsukuba Hospital and its affiliated hospitals, while healthy participants were recruited from the health checkup center of Tsukuba Medical Center Hospital between June 1, 2008, and March 31, 2022 [18]. Some participants were excluded from the study due to missing data regarding BMI, age-onset of asthma or predicted

forced expiratory volume in one second (pFEV1). Ultimately, 1532 healthy individuals and 735 adult asthma patients successfully underwent BMI-GRS calculation and were included in this study (Fig 1A). BMI was defined as weight in kilograms divided by the square of height in meters (kg/m$^2$). Atopy was defined as a positive response (>1.0 lumicount) to at least one of the 14 inhaled allergens [16].

## Ethics statement

This study received approval from the Human Genome Analysis and Epidemiology Research Ethics Committee of the University of Tsukuba (Ethical approval number: H29-294; first approved by the Institutional Review Board on February 21, 2018, and the most recent approval of the amendment on March 28, 2025) as well as from the Human Genome/Gene Analysis Research Ethics Review Committees of the Tsukuba Medical Center. Prior to participation, written informed consent was obtained from each participant in accordance with the principles outlined in the Declaration of Helsinki. To maintain patient anonymity, approved methods were employed as per the Ethics Committee guidelines.

## Genome-wide SNP typing

The genomic DNA of the participants was extracted from peripheral blood samples using the automated DNA extraction system (QuickGene-610L; Fujifilm, Tokyo, Japan). Genome-wide SNP genotyping was carried out using Infinium Asian Screening Array-24 v1.0 BeadChip (Illumina, San Diego, CA, USA) for individuals in Tsukuba Cohort 1 [19], and Illumina HumanHap550v3/610-Quad BeadChip (Illumina) for individuals in Tsukuba Cohort 2 [20].

## SNP selection

Of 85 SNPs associated with BMI (originally identified in a genome-wide association study (GWAS) of the Japanese population [15]), 78 SNPs that showed an association with BMI independent of gender, were initially targeted in the present study. Among these 78, 25 SNPs were identified on the genotyping platforms used for either Tsukuba Cohort 1, Tsukuba Cohort 2, or the additional genetic dataset obtained from the National Bioscience Database Center (NBDC) in Japan, which comprised 565645 SNPs across 112889 individuals (accession number JGAS000114, dataset version hum0014. v6.158k.v1). For the remaining 53 SNPs, alternative SNPs in a strong linkage disequilibrium with the original SNPs

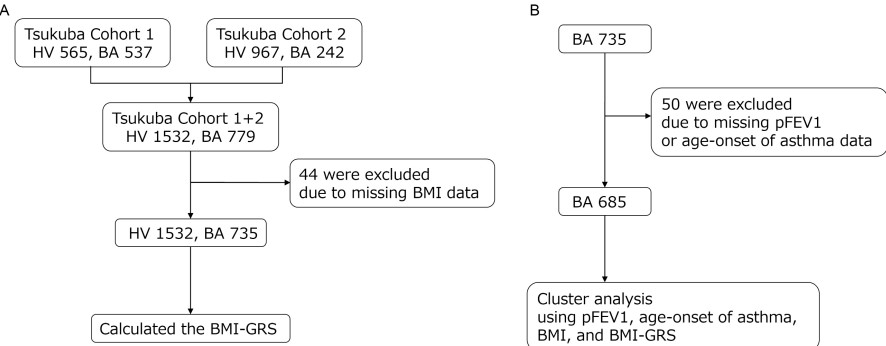

**Fig 1. Study population and data processing workflow.** Participants were recruited from Tsukuba Cohorts 1 and 2, including healthy volunteers and bronchial asthma patients from the University of Tsukuba Hospital and affiliated centers. Healthy volunteers were enrolled through routine health check-ups, while bronchial asthma patients were clinically diagnosed according to established guidelines. Data processing involved excluding participants with incomplete data on BMI, age of asthma onset, or predicted forced expiratory volume in one second. After calculating BMI-genetic risk scores, 685 bronchial asthma patients were included in the final analysis. Cluster analysis was performed using BMI, BMI-genetic risk scores, predicted forced expiratory volume in one second, and age of asthma onset. HV: Healthy Volunteer; BA: Bronchial Asthma; BMI: Body Mass Index; pFEV1: Predicted Forced Expiratory Volume in One Second; BMI-GRS: Body Mass Index-Genetic Risk Score.

($R^2 > 0.8$) were searched for using HaploReg version 4 (https://pubs.broadinstitute.org/mammals/haploreg/haploreg.php). Among these, alternative SNPs were successfully identified for 40 SNPs in at least one of three platforms (Fig 2).

For SNPs that are only present in either Cohort 1, Cohort 2, or the NBDC platform, genotype imputation was performed using a reference panel constructed from the alternate cohort's data or additional genetic information from the NBDC in Japan. The imputation strategy involved two steps: initial pre-phasing of the target dataset to estimate individual haplotypes using the MACH software [21]. followed by genotype imputation with Minimac3 using the pre-phased haplotypes [22] As a result, genotypes at 65 SNPs were successfully obtained for the subsequent analyses.

## Confirmation of the obesity-related genes in our healthy population (Fig 2)

Multivariate linear regression analysis was conducted for 65 SNPs using PLINK on a cohort of 1532 non-asthmatic healthy individuals, using BMI as the outcome variable. Age, gender, and smoking status were included as covariates in the analysis. Ultimately, six SNPs were confirmed as having an association with BMI including the direction of their effects. Five of these six SNPs were expression quantitative trait loci (eQTLs) of genes that are functionally relevant to obesity and are consistently associated with obesity in non-Japanese populations as well as in a Japanese population (Table 1). The standardized regression coefficients (β) calculated during this analysis were subsequently utilized in the calculation of the BMI-GRS.

## Calculation of Body Mass Index-Genetic Risk Score (BMI-GRS)

Using the obtained genotypes and odds ratios (ORs) for each genotype at six SNPs, the GRS was calculated for 1532 healthy individuals and 735 adult asthma patients. The GRS, a composite measure of the cumulative effects of multiple SNPs, is computed as the weighted sum of the risk allele ORs, according to the following formula:

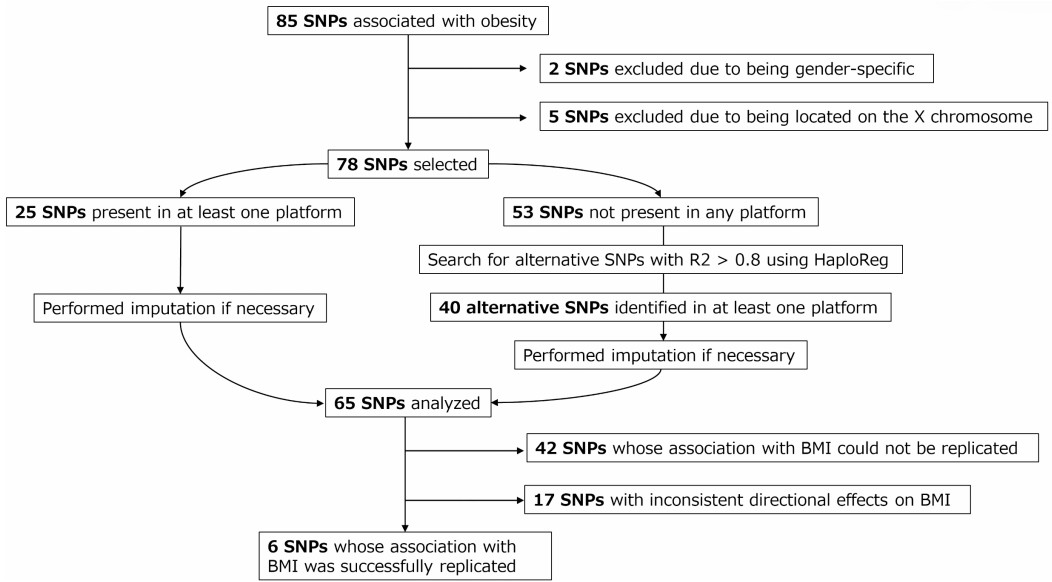

**Fig 2. SNP Selection and genotyping workflow.** This figure summarizes the workflow of SNP selection and genotyping. From 85 BMI-associated SNPs, 78 were initially targeted. After genotype imputation using a reference panel constructed from alternate cohort data and searching for alternative SNPs in strong linkage disequilibrium ($R^2 > 0.8$) using HaploReg, 65 SNPs were retained. Six SNPs with consistent associations with BMI were used for BMI-genetic risk score calculation. SNP: Single Nucleotide Polymorphism; BMI: Body Mass Index; BMI-GRS: Body Mass Index-Genetic Risk Score.

**Table 1. Six SNPs used to calculate BMI – GRS.**

| SNP ID | Alternative SNP[†] | Nearest gene | eQTL | Tissue types | Summary | Reference |
|---|---|---|---|---|---|---|
| rs11642015 | | FTO | FTO | Skeletal muscle | FTO influences body composition and metabolism in mice, affecting fat and lean mass independently of food intake and energy expenditure. The gene was the first gene shown to play a role in common obesity. | [23] |
| rs16937956 | | STK33 | STK33 | Lung, adipose, skeletal muscle, and other tissues | STK33 is linked to childhood obesity. The gene is correlated with higher BMI and obesity rates, suggesting that STK33 may be involved in weight regulation and the development of obesity from an early age. | [24] |
| rs1006257 | rs6882366 ($R^2$ = 0.82) | PCSK1 | ERAP1 | Esophagus-Muscularis and other tissues | The PCSK1 gene is associated with an increased risk of obesity due to reduced enzyme activity. The gene is responsible for obesity in both adults and children, highlighting its important role in polygenic obesity. | [25] |
| rs939584 | rs2867125 ($R^2$ = 0.94) | TMEM18 | ALKAL2 | Adipose-Subcutaneous and other tissues | ALKAL2 plays an important role in the regulation of body weight by affecting energy expenditure and physical activity; ALKAL2 knockout mice show resistance to diet-induced obesity and maintain a lean phenotype, highlighting its importance in metabolic regulation. | [26] |
| rs10208649 | | PSME4 | GPR75 | Lung, adipose, skeletal muscle, and other tissues | The GPR75 gene has been linked to protection against obesity, as rare protein truncating variants in GPR75 are associated with significantly lower BMI and reduced risk of obesity. | [27] |
| rs10197655 | rs11896571 ($R^2$ = 0.87) | LINCO1122 | | | | |

[†]When no original SNPs were identified in any of the 3 cohorts, alternative SNPs in a strong linkage disequilibrium with the original SNPs were searched for (R2 > 0.8) using HaploReg version 4 (https://pubs.broadinstitute.org/mammals/haploreg/haploreg.php).

Abbreviations: SNP, single nucleotide polymorphism; BMI, body mass index; GRS, genetic risk score; eQTL, expression quantitative trait loci. FTO, fat mass and obesity-associated gene; STK33, serine/threonine kinase 33; PCSK1, proprotein convertase subtilisin/kexin type 1; TMEM18, transmembrane protein 18; ALKAL2, ALK and LTK ligand 2; PSME4, proteasome activator subunit 4; GPR75, G protein-coupled receptor 75; LINCO1122, long intergenic non-protein coding RNA 1122.

$$GRSi = \Sigma \beta kRAk$$

where GRSi represents the BMI-GRS for individual *i*; βk denotes the covariance of SNP*k* as the weight of each risk allele derived from our GWAS for BMI, and RAk indicates the number of risk alleles for SNP*k* (0, 1, or 2).

## Statistical analysis

Correlation between BMI-GRS and BMI was examined the in both non-asthmatic healthy individuals and asthmatic patients by calculating Pearson correlation coefficients. Additionally, independent samples t-tests were utilized to investigate whether there were differences in BMI-GRS between non-asthmatic healthy individuals and adult asthmatic patients. Analysis of variance (ANOVA) was used to compare the BMI-GRS according to disease status and BMI.

To identify asthma phenotypes influenced by hereditary predisposition to increased BMI, a two-step cluster analysis of patients with adult asthma was conducted. Previously, we conducted a two-step cluster analysis involving 880 Japanese adult asthma patients, which used eight clinical variables (age, gender, smoking status, age of asthma onset, serum total IgE, specific IgE reactivity to common inhalant allergens, baseline %FEV1, and FEV1/FVC ratio) and identified six phenotypic clusters of adult asthma [18]. The two most influential variables distinguishing these clusters were (1) age of asthma onset and (2) %FEV1. Therefore, in the current study, a two-step cluster analysis was performed, incorporating these two variables along with BMI and BMI-GRS. The similarity between each cluster was measured using log likelihood, and the

optimal number of clusters was defined using Bayesian Information Criterion (BIC) values (https://www.ibm.com/docs/en/spss-statistics).

A total of 685 asthmatics were used in the cluster analysis (Fig 1B), and differences in clinical characteristics between the six clusters were compared using ANOVA or the Kruskal-Wallis test, depending on the data distribution. Clinical characteristics were also compared between the two overweight/obese asthma clusters (Cluster-5 and Cluster-6) using a Bonferroni post-hoc test.

Multinomial logistic regression analysis was conducted to assess the association between BMI genetic risk scores (BMI-GRS) and adult asthma clusters compared to healthy controls. The dependent variable was six asthma clusters (Clusters 1–6) and healthy controls. Healthy controls were designated as the reference category. The primary independent variable was the continuous BMI-GRS, standardized (z-score transformation) to facilitate interpretation as effects per one standard deviation increase. The model was adjusted for sex, age, smoking history, and BMI as potential confounders. Statistical significance was set at $p < 0.05$, with recognition that multiple comparisons across six clusters warrant cautious interpretation of borderline significant results. Potential for circularity bias must be acknowledged, as BMI-GRS was one of the variables used in the original cluster definition. Therefore, results should be interpreted as a descriptive characterization of genetic heterogeneity across the identified phenotypes rather than predictive or causal relationships.

All statistical analyses were performed using SPSS Statistics, version 29.0.0.0 (IBM Corp., Armonk, NY, USA), and p values less than 0.05 were considered as indicating statistical significance.

## Results

In non-asthmatic healthy individuals, the BMI-GRS showed a significant association with BMI ($r^2 = 0.029$, $p = 2.77 \times 10^{-11}$) (Fig 3A). However, in adult patients with asthma, there was no significant correlation between BMI and BMI-GRS ($r^2 = 0.005$; Fig 3B). Additionally, patients with asthma had significantly higher BMI values compared to non-asthmatic healthy controls ($p = 0.002$; Fig 4A), while there was no significant difference in BMI-GRS between the two groups ($p = 0.56$; Fig 4B, Table 2). Absence of an association between BMI-GRS and asthma was confirmed when the comparison was adjusted for BMI ($p = 0.87$).

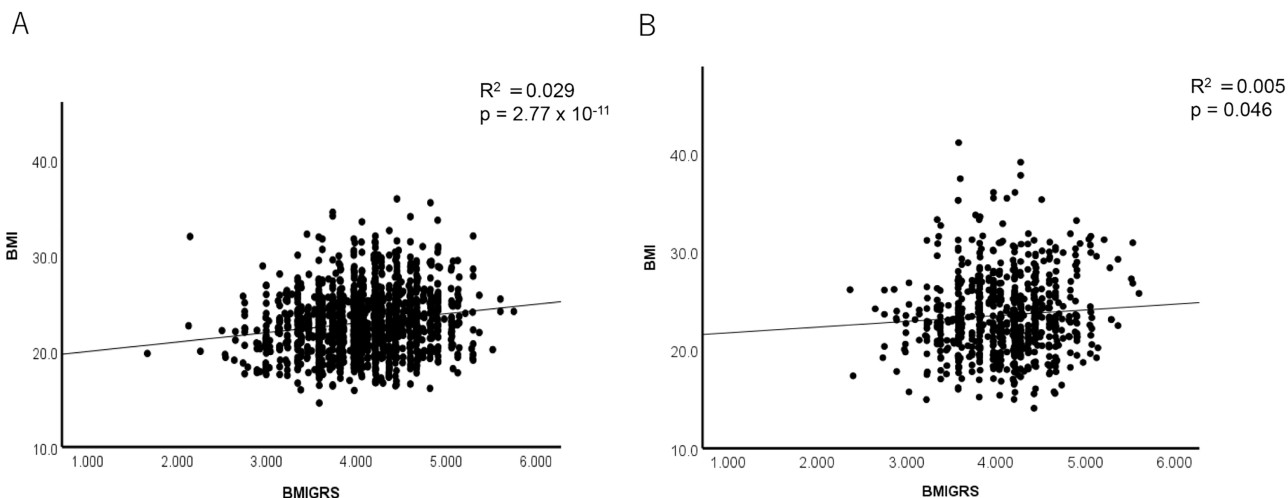

**Fig 3. Correlation between BMI-GRS and BMI in study groups. (A)** A significant positive correlation between BMI and BMI-GRS was observed in non-asthmatic healthy individuals ($R^2 = 0.029$, $p = 2.77 \times 10^{-11}$). **(B)** No significant correlation was found in asthmatic patients ($R^2 = 0.005$, $p = 0.046$), suggesting a limited genetic contribution to BMI in this group. BMI: Body Mass Index; BMI-GRS: Body Mass Index-Genetic Risk Score.

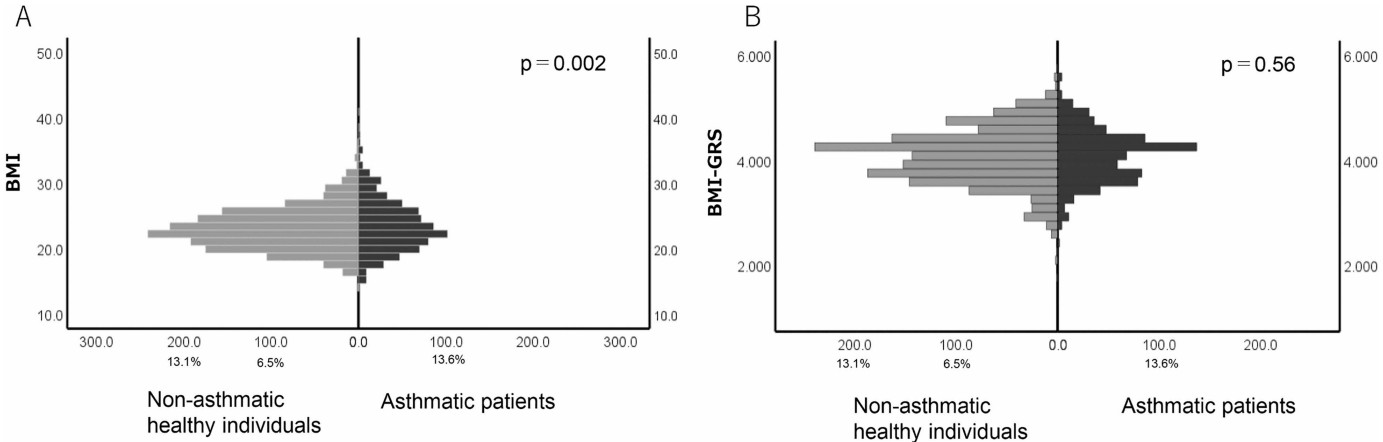

**Fig 4. Comparative analysis of BMI and BMI-GRS between groups. (A)** Asthmatic patients exhibited significantly higher BMI than healthy individuals (p = 0.002). **(B)** No significant difference in BMI-GRS was found between the two groups (p = 0.56), indicating that increased BMI in asthmatic patients is not solely attributable to genetic predisposition. BMI: Body Mass Index; BMI-GRS: Body Mass Index-Genetic Risk Score.

**Table 2. Clinical characteristics of the study participants.**

|  | Healthy individuals | Asthmatic patients | *P value |
|---|---|---|---|
| Number of participants | 1532 | 735 |  |
| Sex (female, %) | 797(52%) | 417(57%) | p = 0.035 |
| Age (yrs) | 51(22-78) | 61(19-90) | p < 0.001 |
| BMI (kg/m2) | 23.05 ± 3.11 | 23.57 ± 4.02 | p < 0.001 |
| BMI ≥ 25, n(%) | 381(25%) | 239(33%) | p < 0.001 |
| BMI ≥ 30, n (%) | 40(3%) | 54(7%) | p < 0.001 |
| BMI-GRS | 4.08 ± 0.53 | 4.09 ± 0.51 | p = 0.563 |
| Smoking (pack-years) |  |  | p = 0.002 |
| 0(%) | 859(56%) | 444(61%) |  |
| 0-10(%) | 203(16%) | 116(11%) |  |
| >10(%) | 468(44%) | 170(18%) |  |
| Atopy (%) | 793(58%) | 428(69%) | p < 0.001 |
| log IgE (IU/mL) | 1.80 ± 0.60 | 2.22 ± 0.64 | p < 0.001 |
| pFEV1 (%) | 92.12 ± 12.79 | 84.38 ± 23.14 | p < 0.001 |
| FEV1/FVC (%) | 82.50 ± 5.83 | 70.82 ± 12.89 | p < 0.001 |

*Differences in clinical characteristics between the 2 groups were compared using unpaired t-test or Mann-Whitney test, depending on the distribution of the data. Categorical variables were compared using the chi-square test.

ANOVA comparisons of BMI-GRS according to disease status and BMI (Fig 5) revealed that the obese control group had the highest BMI-GRS, while the non-obese control group had the lowest. No significant difference in BMI-GRS was noted between non-obese asthma and obese asthma (corrected-*p* = 0.56).

We identified six clusters with distinct clinical profiles (Fig 6). Cluster 1 was characterized by early-onset asthma with the longest disease duration and a lean body habitus; despite their prolonged disease course, these patients maintained relatively preserved lung function. Cluster 2 represented older adult patients with a substantial smoking history who exhibited the most severe airflow limitation, while their BMI and genetic risk for obesity remained moderate. Cluster 3 also

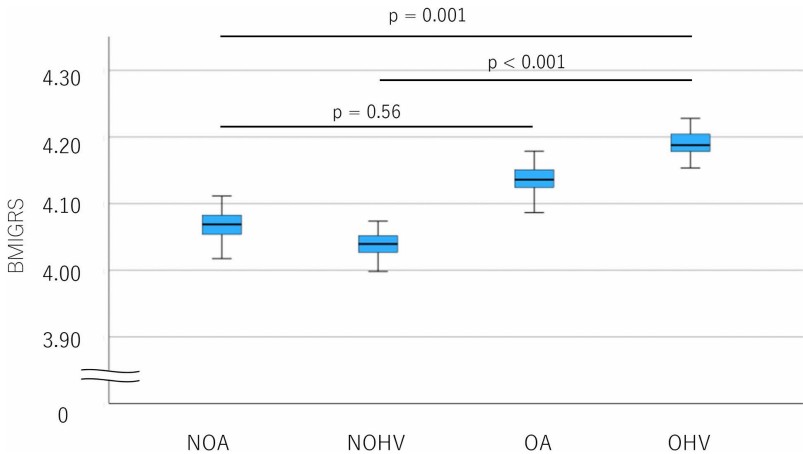

**Fig 5. BMI-GRS comparisons according to disease status and BMI.** Overall group differences were significant (ANCOVA, p<0.001). Post hoc pairwise comparisons (Bonferroni-adjusted) showed that Obese Healthy Volunteers had significantly higher BMIGRS compared with Non-Obese Healthy Volunteers (p<0.001) and Non-Obese Asthma (p=0.001). In contrast, Obese Asthma did not significantly differ from any other group (all p>0.05), including Non-Obese Asthma (p=0.56). No significant differences were observed between asthma and non-asthma groups within the same obesity category. Data are presented as mean±standard error. Statistical differences were evaluated using ANCOVA adjusted for age, sex, and smoking index, followed by Bonferroni-corrected pairwise comparisons. NOA: Non-Obese Asthma, NOHV: Non-Obese Healthy Volunteers, OA: Obese Asthma, OHV: Obese Healthy Volunteers.

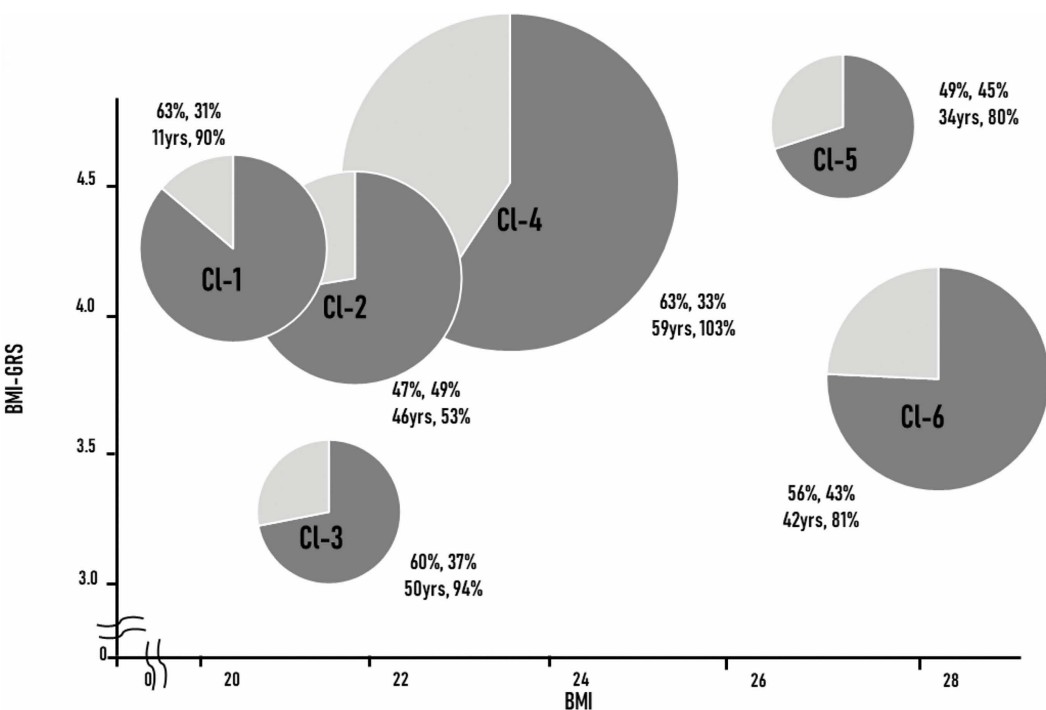

**Fig 6. Summary of cluster analysis of adult asthma.** The six clusters are plotted according to BMI and BMI-GRS. The diameter of each pie represents the population size of each cluster. The grey area of the pie chart indicates the proportion of atopic individuals. In addition, the four numbers given for each pie chart represent clockwise from top left, the percentage of females, the percentage of current and former smokers, pFEV1, and age at asthma onset. BMI: Body Mass Index, BMI-GRS: Body Mass Index-Genetic Risk Score, pFEV1: predicted Forced Expiratory Volume in 1 second.

included older adult patients but was distinguished by the lowest BMI-GRS, indicating minimal genetic predisposition to obesity; these individuals were relatively lean and preserved good lung function. In contrast, Cluster 4 comprised patients with the latest asthma onset and higher BMI-GRS, reflecting a strong genetic tendency toward obesity, yet they maintained preserved pulmonary function. Cluster 5 was defined by obesity combined with the highest BMI-GRS, representing individuals with both a phenotypic and genetic predisposition to obesity and showing moderate airflow limitation. Finally, Cluster 6 also displayed an obese phenotype but was associated with relatively low BMI-GRS, suggesting that non-genetic factors may predominantly underlie obesity in this group; their lung function was moderately impaired.

A comparison of clinical characteristics between two obese/overweight clusters using the Bonferroni post hoc test showed that there were differences only in disease duration between the two clusters, but not in age of asthma onset, type 2 markers (serum total IgE levels, peripheral blood eosinophil count, FeNO), or airflow obstruction (Table 3).

We conducted multivariable logistic regression modeling comparing BMI-GRS contribution to each cluster adjusting for age, sex, smoking history and BMI, explicitly framed as descriptive evidence for genetic heterogeneity rather than predictive modeling. We found significant associations between BMI-GRS and four of the six asthma clusters (Table 4). Compared to healthy controls, Cluster-3 (lower BMI) demonstrated significantly reduced BMI-GRS, while Cluster-4 (normal BMI) showed significantly elevated BMI-GRS. Among the overweight/obese clusters, Cluster-5 exhibited significantly elevated BMI-GRS, whereas Cluster-6 demonstrated significantly reduced BMI-GRS. Clusters 1 and 2 showed no significant differences compared to healthy controls ($p > 0.05$ for both).

**Table 3. Detailed characteristics of each asthma cluster.**

| | Cluster-1 | Cluster-2 | Cluster-3 | Cluster-4 | Cluster-5 | Cluster-6 | *P value |
|---|---|---|---|---|---|---|---|
| Number of participants | 108 | 116 | 83 | 174 | 83 | 121 | |
| Sex (female, %) | 68 (63%) | 55 (47%) | 50 (60%) | 109 (63%) | 41 (49%) | 68 (56%) | p > 0.05 |
| Age (yrs) | 38.5 (19-80) | 66.5 (24-90) | 64 (21-87) | 66 (31-89) | 57 (23-86) | 60 (19-88) | p < 0.001 |
| BMI (kg/m2) | 20.7±2.2 | 21.4±2.8 | 21.8±2.3 | 23.0±3.0 | 27.2±3.0 | 28.2±3.6 | p < 0.001 |
| BMI-GRS | 4.10±0.31 | 4.08±0.41 | 3.33±0.31 | 4.37±0.31 | 4.69±0.33 | 3.78±0.26 | p < 0.001 |
| Age-onset (yrs) | 11 (0-49) | 45.5 (2-82) | 50 (1-82) | 59 (29-87) | 34 (1-70) | 42 (1-82) | p < 0.001 |
| Duration of asthma (yrs) | 24 (0-73) | 17 (0-67) | 6 (0-60) | 8 (0-34) | 21 (0-65) | 14 (0-64) | p < 0.001 |
| Pack-years | | | | | | | p = 0.004 |
| :0 (%) | 69.4 | 50.9 | 63.4 | 67.2 | 55.4 | 56.7 | |
| :0-10 | 20.4 | 17.2 | 12.2 | 14.4 | 21.7 | 13.3 | |
| :>10 | 10.2 | 31.9 | 24.4 | 18.4 | 22.9 | 30 | |
| Atopy (%) | 81 (86%) | 76 (72%) | 44 (62%) | 92 (59%) | 49 (70%) | 78 (76%) | p < 0.001 |
| Blood eosinophil count (/μL) | 280.5 (0-1848) | 274 (0-1819) | 310 (0-3354) | 210.56 (0-5096) | 260 (0-1320) | 297.5 (0-2360) | p > 0.05 |
| log IgE (IU/mL) | 2.44±0.39 | 2.22±0.64 | 2.17±0.66 | 2.12±0.64 | 2.14±0.63 | 2.22±0.58 | p = 0.003 |
| FeNO (ppb) | 26 (6-126) | 31 (2-156) | 22 (0-300) | 25 (5-160) | 20 (4-82) | 33.5 (1-177) | p > 0.05 |
| pFEV1 (%) | 89.9±13.5 | 52.8±12.8 | 94±17.4 | 103.1±16.8 | 79.8±17.5 | 80.6±17.0 | p < 0.001 |
| FEV1/FVC (%) | 77.3±10.0 | 56.7±12.6 | 74.8±10.5 | 75.2±8.8 | 70.4±12.5 | 71.6±10.5 | p < 0.001 |

Data for BMI, BMI-GRS, logIgE, pFEV1, and FEV1/FVC are presented as mean±SD. Age, age of onset, duration of asthma, blood eosinophil count, and FeNO are presented as median (range). BMI: Body Mass Index, BMI-GRS: Body Mass Index – Genetic Risk Score, pFEV1: predicted Forced Expiratory Volume in 1 second, FEV1/FVC: Forced Expiratory Volume in 1 second/ Forced Vital Capacity, FeNO: Fractional exhaled Nitric Oxide

We defined atopy as a positive IgE response to at least 1 of 14 common inhaled allergens.

*Differences in clinical characteristics between the 6 clusters were compared using ANOVA or Kruskal-Wallis test, depending on the distribution of the data.

**Table 4. Multinominal logistic regression model of BMI-GRS on adult asthma clusters.**

| Cluster | Cluster-1 | Cluster-2 | Cluster-3 | Cluster-4 | Cluster-5 | Cluster-6 |
|---|---|---|---|---|---|---|
| β (SE) | 0.07 (0.05) | 0.04 (0.05) | −0.73 (0.05) | 0.29 (0.04) | 0.52 (0.06) | −0.42 (0.05) |
| 95% CI | −0.03 to 0.17 | −0.06 to 0.13 | −0.83 to −0.62 | 0.21 to 0.37 | 0.41 to 0.63 | −0.51 to −0.32 |
| p value | 0.15 | 0.43 | <0.001 | <0.001 | <0.001 | <0.001 |

Values are presented as regression coefficients (β) with standard errors (SE) and 95% confidence intervals (CI). Reference group: healthy volunteers. Multivariable linear regression model was adjusted for sex, age, smoking index, and BMI. P values <0.05 were considered statistically significant.

## Discussion

This study of Japanese individuals reveals some key findings that challenge conventional understanding of the obesity-asthma relationship. First, while asthmatics had higher BMI than controls (confirming established epidemiological associations), no overall difference in BMI-GRS existed between groups, indicating that genetic predisposition to elevated BMI does not uniformly increase asthma risk. Second, multinomial logistic regression identified significant BMI-GRS differences in four of six asthma clusters compared to healthy controls—reduced BMI-GRS in Clusters 3 (low BMI) and 6 (obese), and elevated BMI-GRS in Clusters 4 (normal BMI) and 5 (obese). Third, this pattern demonstrates genetic heterogeneity spanning the entire BMI spectrum, with distinct etiologic subtypes existing within phenotypically similar groups, particularly among obese patients where genetic predisposition distinguishes between genetically driven (Cluster-5) and environmentally driven (Cluster-6) obesity-related asthma. These findings reveal that BMI-associated genetic variants influence asthma through heterogeneous mechanisms beyond traditional obesity-mediated pathways, supporting precision medicine approaches based on genetic risk profiles rather than BMI classifications alone.

Obesity-to-asthma pathways include mechanical effects (reduced chest wall compliance and lung volumes leading to airway closure and hyperresponsiveness [8,9]), systemic inflammation (elevated TNF-α/IL-6 and altered adipokines with increased ratio of leptin/adiponectin promoting Th1/Th17 and neutrophilic inflammation [7,10]), metabolic dysregulation (increased ADMA reducing nitric oxide bioavailability [11]), hormonal influences (dysregulated sex hormones affecting airway inflammation [12]), and gut-lung axis changes (obesity-related dysbiosis sustaining systemic inflammation [1]). Asthma-to-obesity pathways include reduced physical activity due to exertional dyspnea or fear of exacerbation [7,8], and chronic corticosteroid-induced weight gain through metabolic effects including increased appetite and insulin resistance [28]. Our study revealed critical genetic heterogeneity within obesity-related asthma, identifying two etiologically distinct phenotypes despite similar BMI levels: Cluster-5 with significantly elevated BMI-GRS and Cluster-6 with significantly reduced BMI-GRS. Cluster-5 represents genetically driven obese asthma, while Cluster-6 suggests environmentally driven obese asthma, where obesity results primarily from asthma-to-obesity pathways. This genetic stratification within phenotypically similar obese asthmatics provides novel insights into asthma heterogeneity and supports precision medicine approaches.

The genes contributing to our BMI-GRS possess diverse molecular functions that provide biologically plausible pathways for respiratory effects independent of BMI, potentially involving inflammatory regulation (*FTO*, *ERAP1*), metabolic signaling (*PCSK1*), immune modulation (*PSME4*), and neuroendocrine pathways (*GPR75*), all operating independently of current BMI status [29–32]. The presence of elevated BMI-GRS in non-obese (Cluster-4) asthma patients, therefore, suggested that these genetic effects may modulate the inflammatory, metabolic, and hormonal mechanisms described above, potentially explaining why some individuals with elevated BMI-GRS develop asthma despite normal BMI (genetic predisposition manifesting through non-BMI pathways).

While this study provides valuable new insights, it is important to acknowledge its limitations. One major limitation is the cross-sectional design, which captures only a single point in time, preventing us from establishing a causal relationship between BMI and asthma. For instance, in adults with childhood-onset asthma, it remains unclear whether they were

already overweight or obese when their asthma symptoms first appeared. Future research requires the following: (1) Prospective cohort studies that follow genetically stratified individuals from health to disease onset. (2) Comprehensive multifactorial analyses that control for environmental, lifestyle, and treatment confounders. (3) Mechanistic studies that investigate the biological pathways through which BMI-associated variants influence respiratory outcomes.

In conclusion, this cross-sectional study demonstrates genetic heterogeneity within adult asthma phenotypes in a Japanese population. BMI-associated genetic variants showed differential distribution patterns across six distinct asthma clusters, with elevated BMI-GRS observed in both obese and non-obese phenotypes. These descriptive findings suggest that BMI-associated genetic variants may influence asthma susceptibility through multiple biological pathways. This represents a paradigm shift from reactive phenotype-based medicine to predictive, genetically informed precision medicine, and has the potential to transform the current single "obese asthma" category into a genetically stratified approach, even when clinical findings are similar.

## Supporting Information

**S1 File. Supporting information.csv: Minimal data set underlying the findings of this study.**
(CSV)

## Acknowledgments

The authors thank all the participants in this study. We also thank Ms. Takako Nakamura for technical assistance. We are grateful to the Medical English Communications Center (MECC), University of Tsukuba, for their assistance in English editing.

## Author contributions

**Conceptualization:** Yohei Yatagai, Hiroyuki Oshima, Nobuyuki Hizawa.

**Data curation:** Yohei Yatagai, Hiroyuki Oshima, Yu Abe, Haruna Kitazawa, Hironori Masuko, Takashi Naito, Takefumi Saito, Tohru Sakamoto, Nobuyuki Hizawa.

**Formal analysis:** Yohei Yatagai, Hiroyuki Oshima, Tomomitsu Hirota, Mayumi Tamari, Emiko Noguchi, Nobuyuki Hizawa.

**Investigation:** Yohei Yatagai, Hiroyuki Oshima, Yu Abe, Haruna Kitazawa, Hironori Masuko, Takashi Naito, Takefumi Saito, Tohru Sakamoto, Nobuyuki Hizawa.

**Methodology:** Yohei Yatagai, Hiroyuki Oshima, Nobuyuki Hizawa.

**Supervision:** Nobuyuki Hizawa.

**Writing – original draft:** Yohei Yatagai, Hiroyuki Oshima.

**Writing – review & editing:** Nobuyuki Hizawa.

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
