## [Decision Letter · Decision Letter 0]

7 Aug 2025

Dear Dr.  Yatagai,

Thank you for your email and for the opportunity to revise and resubmit our manuscript titled "Impact of hereditary predisposition to increased BMI on adult asthma phenotype". We appreciate the time and effort the reviewers and editorial team have taken to evaluate our work.

We look forward to receiving your revised manuscript.

Kind regards,

Muhammad Salman Bashir, M.S.C

Academic Editor

PLOS ONE

Journal Requirements:

3. We note that your Data Availability Statement is currently as follows:

“All relevant data are within the manuscript and its Supporting Information files.”

Additional Editor Comments:

We are encouraged by your assessment that the manuscript has merit and are grateful for the constructive feedback provided. In response, we have carefully revised the manuscript to address all the comments and concerns raised during the review process. We believe these revisions have significantly improved the clarity, rigor, and overall quality of the manuscript.

Indicate which changes you require for acceptance versus which changes you recommendAddress any conflicts between the reviews so that it's clear which advice the authors should followProvide specific feedback from your evaluation of the manuscript

publication criteria  and not, for example, on novelty or perceived impact.

**Comments to the Author**

1. Is the manuscript technically sound, and do the data support the conclusions?

Reviewer #1: Partly

Reviewer #2: Yes

Reviewer #3: Partly

Reviewer #4: Partly

2. Has the statistical analysis been performed appropriately and rigorously?

Reviewer #1: Yes

Reviewer #2: I Don't Know

Reviewer #3: Yes

Reviewer #4: I Don't Know

3. Have the authors made all data underlying the findings in their manuscript fully available?

Reviewer #1: No

Reviewer #2: Yes

Reviewer #3: Yes

Reviewer #4: Yes

4. Is the manuscript presented in an intelligible fashion and written in standard English?

Reviewer #1: Yes

Reviewer #2: Yes

Reviewer #3: Yes

Reviewer #4: Yes

Reviewer #1: Line 35 isn't justified based on the finding - suggest remove 'asthma may contribute to increased BMI.'

Comparison should have been made between asthma vs overweight asthma and overweight vs overweight asthma.

Why 2 different cohorts?

Why were different SNP genotyping methods used?

Incomplete summary for rs939584 in table.

Duplication of SNP selection and BMI-GRS score in methods and results.

Line 206-212 should be in discussion

Table 2 - Mann Whitney spelling error. Was there a weighted difference in BMI due to the difference in number of healthy and asthmatic participants for a difference of 0.52 to be significant?

A graph showing no correlation between BMIGRS and asthma should be included

Reviewer #2: This is a comprehensive and interesting study about the association between adult onset asthma and BMI, in a large cohort of subjects. A minör point to critise is the mean BMI of patients and healthy controls is not matched! It is better to analyse at least a subgroup of controls with a matched BMI with the asthmatics.

Reviewer #3: The study aimed to investigate the role of genetic predisposition to increased BMI, evaluated by calculating a BMI-genetic risk score (BMI-GRS), on adult asthma phenotypes and explore whether these genetic factors might distinguish certain subgroups of overweight or obese asthma patients. The study included 1530 healthy individuals and 735 adult asthma patients. Six single nucleotide polymorphisms (SNPs) which were confirmed as having an association with BMI in the healthy population in the study were used to calculate BMI-GRS. The authors found that patients with asthma had significantly higher BMI values compared to non-asthmatic healthy controls, but there was no significant difference in BMI-GRS between the two groups. The study reported that no significant correlation was found between BMI and BMI-genetic risk scores in asthmatic patients. The authors performed a cluster analysis using age of asthma onset, %FEV1, BMI and BMI-GRS. This analysis identified 6 distinct asthma patient groups, among these, two clusters were characterized by increased BMI, one with a higher BMI-GRS and the other with a lower BMI-GRS. I have some major and minor comments.

- Lines 174-178, the population attributable risk fraction (PARF) was calculated assuming there are no confounding factors, and no confounding adjustment in the calculation of PARF was performed. Additionally, PARF assumes that a cause-effect relationship exists between the exposure and outcome. However, the study reported that there was no significant correlation between BMI and BMI-genetic risk scores in asthmatic patients. This is a limitation.

- Abbreviations should be added to Table 1. In table 1 the summary part of rs939584 is not completed.

- Lines 187-199 of the results section is about methods.

- Line 197, table 1a, 1b, there is no table 1a or 1b.

- Lines 206-212 of results section would be more appropriate in discussion.

- I suggest adding more clinical characteristics on table 2, like BMI groups, obesity and overweight rates.

- I suggest adding the criteria defining atopy to the methods section.

- Lines 229-235 of the results section is about methods.

- The authors stated in lines 268-270 that ‘Our results suggest that genetic factors influencing BMI may play a crucial role in the development of adult-onset overweight/obese asthma in certain patients, particularly those in Cluster-5, where a higher BMI-genetic risk score (BMI-GRS) was observed.’. For this comment multivariate analyses should be performed. Additionally, no significant correlation was found between BMI and BMI-genetic risk scores in asthmatic patients in the study.

- Lines 276-285, If clinical features are not different among the two phenotypes (cluster 5 and 6), what is the added value of BMI-GRS calculation? I suggest discussing this.

- I suggest adding clinical comparison of all 6 clusters to the results section.

- In lines 294-296, it is stated that ‘Specifically, asthma symptoms may lead to weight gain in some patients due to reduced physical activity or medication side effects, while genetic predispositions to obesity may increase the risk of developing asthma in others.’. I suggest adding previous literature data and a detailed discussion.

- In lines 297-299, it is stated that ‘In conclusion, this study supports the growing body of evidence suggesting that genetic susceptibility to higher BMI is a key factor in the development of adult-onset asthma in certain subgroups of overweight or obese patients.’. As the study does not include a multifactorial analysis and a longitudinal analysis, I suggest to revise the conclusion part to be supported by the data.

- The discussion section of the manuscript includes only one reference. The discussion section needs a revision and expansion to discuss the results with other previous studies.

- I suggest giving percentages in the x axes of figure 4.

Reviewer #4: I read with interest the manuscript entitled “Impact of hereditary predisposition to increased BMI on adult asthma phenotype”. The manuscript describes the results of a study delving into the genetic susceptibility to elevated body mass index (BMI) and its subsequent influence on adult-onset asthma, specifically within a Japanese cohort. This study indicates a bidirectional relationship: genetic predisposition to a higher BMI appears to contribute to adult-onset overweight/obese asthma, and conversely, asthma itself may lead to increased BMI. Therefore, the study provides valuable results that contribute to progress in the fields of precision medicine and precision nutrition.

The manuscript is generally well-written. Nevertheless, some aspects of the manuscript should be addressed before a decision on publication. The manuscript needs revision focused on the remarks stated below.

MAJOR REMARKS:

• Title: Please consider reformulating the title so that it is understood that the study was conducted in Japan. This suggestion is made in the context that the Japanese population has particularities regarding being classified as overweight/obese in relation to BMI.

• Line 42: Please provide recent data regarding the prevalence of both obesity and asthma, at global level and in Japan, respectively.

• Line 45: Please define BMI.

• Lines 45-49: The mechanisms by which obesity contributes to the pathogenesis of asthma need to be described more extensively, citing a significantly larger number of scientific publications. For instance, potential mediators between obesity and asthma should be briefly described (e.g., mechanical changes, systemic inflammation, sex hormones, arginine metabolism, gut microbiome) (https://pmc.ncbi.nlm.nih.gov/articles/PMC9671155/#S3;
https://www.sciencedirect.com/science/article/pii/S1323893023000837?via%3Dihub#sec3). Asthma phenotypes in obesity (e.g., Late-Onset Asthma

Phenotype, Early-Onset Asthma

Phenotype) should also be mentioned (https://pmc.ncbi.nlm.nih.gov/articles/PMC11204497/#sec5-jcm-13-03474).

• Discussion section: this section must be thoroughly revised. Discussions must be based on scientific articles already published on the topic of the study. Currently, only one scientific article is cited in the discussion section, and this aspect needs to be significantly improved.

MINOR REMARKS:

• Line 92: Please provide the date of issue of the ethical approval number.

• Table 1: The references in the table should be moved to the reference list and kept in the table only as numbers.

• Figure 5: Please remove the text “Figure 1” from this figure.

**Do you want your identity to be public for this peer review?** For information about this choice, including consent withdrawal, please see our Privacy Policy

Reviewer #1: No

Reviewer #2: No

Reviewer #3: No

Reviewer #4: No

---

## [Author Response · Author response to Decision Letter 1]

29 Sep 2025

Reviewer #1

Comment #1: Line 35 isn't justified based on the finding - suggest remove 'asthma may contribute to increased BMI.'

Response #1: Based on the comments, we have revised line 35 of the abstract. The revised version reads: “This study demonstrates a genetic heterogeneity in the phenotype of adult asthma among a Japanese population, showing that genetic variants associated with BMI contribute to specific subtypes of asthma.”

In addition, the following revisions were made to adjust the future research direction and ensure consistency throughout the abstract, focusing on a longitudinal approach to understand temporal relationships rather than assuming bidirectional causality.

“Prospective longitudinal studies are essential to delineate the interactions between genetic predisposition, elevated BMI, subsequent changes in adiposity, and the evolution of asthma phenotypes, which would facilitate the development of mechanism-based therapeutic strategies tailored to genetically-defined patient subgroups.”

Comment #2: Comparison should have been made between asthma vs overweight asthma and overweight vs overweight asthma.

Response #2: In the revised manuscript, new Figure 5 shows a comparison of BMI-GRS between four groups: non-obese controls, obese controls, non-obese asthmatics and obese asthmatics. The obese control group had the highest BMI-GRS, while the non-obese control group had the lowest. No significant difference in BMI-GRS was noted between non-obese asthma and obese asthma (corrected p= 0.56).

Comment #3: Why 2 different cohorts?

Response #3: Our sequential recruitment strategy (Tsukuba Cohort 2: 2008–2012; Tsukuba Cohort 1: 2013–2022) was designed to maximize the number of participants and thereby maximize the statistical power and scientific rigor while addressing practical and ethical considerations (PLoS One 2024;19(3):e0300000. doi:10.1371/journal.pone.0300000.). The decade-long recruitment period and consistent geographic focus on Ibaraki Prefecture enhance representativeness while maintaining genetic homogeneity within the study’s Japanese population. However, recruiting over 2,000 participants with comprehensive clinical and genetic data within a short time frame presents significant ethical and logistical challenges. Our phased approach enabled continuous patient enrollment from Tsukuba University Hospital and its affiliated institutions.

Comment #4: Why were different SNP genotyping methods used?

Response #4: The evolution of genotyping technology during our extended collection period necessitated platform updates. For Tsukuba Cohort 2, Illumina HumanHap 550K/610-Quad BeadChips provided comprehensive, genome-wide coverage, representing the gold standard from 2008 to 2012. For Tsukuba Cohort 1, the Infinium Asian Screening Array-24 v1.0 was optimized specifically for East Asian populations and incorporated population-specific variants with enhanced coverage of Asian-specific alleles.

Comment #5: Incomplete summary for rs939584 in table.

Response #5: We have corrected the summary for rs939584 in Table 1.

Comment #6: Duplication of SNP selection and BMI-GRS score in methods and results.

Response #6: Thank you for identifying this methodological redundancy. We have addressed this duplication by removing the detailed description of SNP selection and BMI-GRS calculation from the Results section while retaining the comprehensive methodological details in the Methods section.

Comment #7: Line 206-212 should be in discussion

Response #7: According to the comment, we have moved these sentences to discussion Line 305 to 308 in the revised manuscript.

Comment #8: Table 2 - Mann Whitney spelling error. Was there a weighted difference in BMI due to the difference in number of healthy and asthmatic participants for a difference of 0.52 to be significant?

Response #8: The spelling mistake the reviewer pointed out has been corrected.

We believe that the significance is not due to an artefact of "weighted difference" from unequal sample sizes. The large, combined sample size (n = 2,267) increased statistical power and reduced the standard error, enabling the precise detection of a small but genuine difference. While this 0.52 BMI difference is statistically significant, its clinical significance should be evaluated separately from the statistical result.

Comment #9: A graph showing no correlation between BMIGRS and asthma should be included.

Response #9: Thank you for this suggestion. Figure 4B shows no correlation between BMI-GRS and asthma. We confirmed that asthma is not associated with the BMI-GRS, even when the analysis was adjusted for BMI (p = 0.87).

Reviewer #2:

Comment #1: This is a comprehensive and interesting study about the association between adult onset asthma and BMI, in a large cohort of subjects. A minör point to critise is the mean BMI of patients and healthy controls is not matched! It is better to analyse at least a subgroup of controls with a matched BMI with the asthmatics.

Response #1: We appreciate your thoughtful suggestion regarding BMI matching. However, we avoided this approach in our study design for several critical methodological reasons. First, our BMI-GRS formula was constructed using the entire healthy cohort (n = 1,532) to maximize statistical power and generalizability to the Japanese population. Of the 85 previously identified BMI-associated SNPs, only six showed significant associations in our cohort. Restricting the discovery sample further through BMI matching would substantially increase the risk of false negatives and compromise the validity of the genetic risk score. Second, the two-phase approach (discovery in healthy individuals and application to disease cohorts) is the gold standard in genetic epidemiology. This approach ensures that genetic risk scores reflect population-level genetic architecture rather than disease-specific patterns. Restricting the BMI range in our discovery cohort would introduce systematic bias in genetic effect estimation and reduce the generalizability of our findings to the broader Japanese population.

However, to address the reviewer’s concern without compromising our BMI-GRS construction, we conducted BMI-adjusted models to estimate the direct effects of the BMI-GRS by comparing non-asthmatic healthy controls with patients with asthma. This analysis confirmed that asthma is not associated with the BMI-GRS, even when adjusted for BMI (p = 0.87). We have mentioned the result in the revised manuscript (Line 220 to 222).

Reviewer #3:

The study aimed to investigate the role of genetic predisposition to increased BMI, evaluated by calculating a BMI-genetic risk score (BMI-GRS), on adult asthma phenotypes and explore whether these genetic factors might distinguish certain subgroups of overweight or obese asthma patients. The study included 1530 healthy individuals and 735 adult asthma patients. Six single nucleotide polymorphisms (SNPs) which were confirmed as having an association with BMI in the healthy population in the study were used to calculate BMI-GRS. The authors found that patients with asthma had significantly higher BMI values compared to non-asthmatic healthy controls, but there was no significant difference in BMI-GRS between the two groups. The study reported that no significant correlation was found between BMI and BMI-genetic risk scores in asthmatic patients. The authors performed a cluster analysis using age of asthma onset, %FEV1, BMI and BMI-GRS. This analysis identified 6 distinct asthma patient groups, among these, two clusters were characterized by increased BMI, one with a higher BMI-GRS and the other with a lower BMI-GRS. I have some major and minor comments.

Comment #1: Lines 174-178, the population attributable risk fraction (PARF) was calculated assuming there are no confounding factors, and no confounding adjustment in the calculation of PARF was performed. Additionally, PARF assumes that a cause-effect relationship exists between exposure and outcome. However, the study reported that there was no significant correlation between BMI and BMI-genetic risk scores in asthmatic patients. This is a limitation.

Response #1: Thank you for raising these important methodological concerns about our PARF analysis. After careful consideration of the reviewer feedback and our study's key findings, we have decided to remove the PARF analysis entirely from the manuscript. The lack of correlation between BMI-GRS and BMI in asthmatic patients violates the basic causal assumptions required for valid PARF interpretation. In fact, the presence of high BMI-GRS cluster among non-obese asthmatic patients fundamentally challenges the traditional mediation pathway (BMI-GRS → BMI → obesity-related asthma) that underlies PARF calculations. We now consider this finding indicates that BMI-associated genetic variants influence asthma through multiple biological pathways, including both BMI-associated and BMI-independent mechanisms.

In the revised manuscript, we have replaced PARF with a multinomial logistic regression model (7 categories including healthy controls) to quantify genetic risk profiles across asthma phenotypes that provide a more accurate and comprehensive assessment of genetic heterogeneity in asthma pathogenesis while avoiding the methodological limitations inherent in PARF calculations under our study conditions.

Comment #2: - Abbreviations should be added to Table 1. In table 1 the summary part of rs939584 is not completed.

Response #2: We have added the relevant abbreviations as a footnote to Table 1. We have also corrected the summary for rs939584 in Table 1.

Comment #3: - Lines 187-199 of the results section is about methods.

Response #3: Thank you for identifying this methodological redundancy. We have addressed this duplication by removing the detailed description of SNP selection and BMI-GRS calculation from the Results section while retaining the comprehensive methodological details in the Methods section.

Comment #4: - Line 197, table 1a, 1b, there is no table 1a or 1b.

Response #4: As you correctly noted, there is only Table 1, and it was mistakenly referred to as Table 1A and 1B. We have corrected the text so that it consistently refers to Table 1.

Comment #5: - Lines 206-212 of results section would be more appropriate in discussion.

Response #5: According to the comment, we have moved these sentences to discussion (line 305 to 308) in the revised manuscript.

Comment #6: - I suggest adding more clinical characteristics on table 2, like BMI groups, obesity and overweight rates.

Response #6: In accordance with your comment, we have added additional clinical characteristics to Table 2, including the number and percentage of participants with BMI ≥25 and BMI ≥30 in both the healthy control and asthma groups.

Comment #7: - I suggest adding the criteria defining atopy to the methods section.

Response #7: The definition of atopy has been added to the Study participants section of Material and Methods.

Comment #8: - Lines 229-235 of the results section is about methods.

Response #8: We have moved the sentences to the methods section in the revised manuscript.

Comment #9: - The authors stated in lines 268-270 that ‘Our results suggest that genetic factors influencing BMI may play a crucial role in the development of adult-onset overweight/obese asthma in certain patients, particularly those in Cluster-5, where a higher BMI-genetic risk score (BMI-GRS) was observed.’. For this comment multivariate analyses should be performed. Additionally, no significant correlation was found between BMI and BMI-genetic risk scores in asthmatic patients in the study.

Response #9: We sincerely appreciate this crucial methodological critique, which has prompted us to fundamentally reconsider our analytical approach and transform our interpretation of key findings. Rather than viewing the lack of BMI-GRS/BMI correlation in asthmatic patients as problematic, we now recognize this as a pivotal discovery providing direct evidence for genetic heterogeneity and pathway diversity in asthma pathogenesis.

We conducted multivariable logistic regression modeling comparing BMI-GRS contribution to each cluster adjusting for age, sex, smoking history and BMI, explicitly framed as descriptive evidence for genetic heterogeneity rather than predictive modeling. Among the six distinct asthma phenotypes identified, we found two overweight/obese clusters (one with elevated BMI-GRS, one without) and four non-obese clusters (with one showing significantly elevated BMI-GRS). This pattern demonstrates that BMI-associated genetic variants contribute to asthma susceptibility through heterogeneous biological mechanisms spanning the entire BMI spectrum - both BMI-mediated pathways (traditional obesity-related mechanisms) and BMI-independent pleiotropic pathways.

We have reframed our conclusions from causal prediction to evidence-based genetic heterogeneity assessment. "BMI-GRS plays a crucial role in development of Cluster-5 asthma" was revised to read " BMI-associated genetic variants showed differential distribution patterns across six distinct asthma clusters, with elevated BMI-GRS observed in both obese and non-obese phenotypes. These descriptive findings suggest that BMI-associated genetic variants may influence asthma susceptibility through multiple biological pathways. "

Comment #10: - Lines 276-285, If clinical features are not different among the two phenotypes (cluster 5 and 6), what is the added value of BMI-GRS calculation? I suggest discussing this.

Response #10: Thank you for this suggestion. We believe that the added value of BMI-GRS lies not in identifying current different phenotypes, but in revealing unique pathogenic mechanisms that may be useful for prognosis, treatment selection, and risk stratification. This represents a paradigm shift from reactive phenotype-based medicine to predictive, genetically informed precision medicine, and has the potential to transform the current single ‘obese asthma’ category into a genetically stratified approach, even when clinical findings are similar, thereby having a concrete impact on personalized patient care. This issue is discussed in the revised manuscript.

Comment #11: - I suggest adding clinical comparison of all 6 clusters to the results section.

Response #11: Thank you. Based on your suggestion, we have added clinical comparison of all 6 clusters to the result section as follows: We identified six clusters with distinct clinical profiles. Cluster 1 was characterized by early-onset asthma with the longest disease duration and a lean body habitus; despite their prolonged disease course, these patients maintained relatively preserved lung function. Cluster 2 represented elderly patients with a substantial smoking history who exhibited the most severe airflow limitation, while their BMI and genetic risk for obesity remained moderate. Cluster 3 also included elderly patients but was distinguished by the lowest BMI-GRS, indicating minimal genetic predisposition to obesity; these individuals were relatively lean and preserved good lung function. In contrast, Cluster 4 comprised patients with the latest asthma onset and higher BMI-GRS, reflecting a strong genetic tendency toward obesity, yet they maintained preserved pulmonary function. Cluster 5 was defined by obesity combined with the highest BMI-GRS, representing individuals with both a phenotypic and genetic predisposition to obesity and showing moderate airflow limitation. Finally, Cluster 6 also displayed an obese phenotype but was associated with relatively low BMI-GRS, suggesting that non-genetic factors may predominantly underlie obesity in this group; their lung function was moderately impaired.

Comment #12: - In lines 294-296, it is stated that ‘Specifically, asthma symptoms may lead to weight gain in some patients due to reduced physical activity or medication side effects, while genetic predispositions to obesity may increase the risk of developing asthma in others.’. I suggest adding previous literature data and a detailed discussion.

Response#12: Thank you for your proposal. The association between asthma and obesity is increasingly recognized as a bidirectional and mechanistical

---

## [Decision Letter · Decision Letter 1]

26 Dec 2025

Genetic predisposition to elevated BMI and adult asthma phenotypes in a Japanese population

PONE-D-25-02383R1

Dear Dr. Yatagai,

We’re pleased to inform you that your manuscript has been judged scientifically suitable for publication and will be formally accepted for publication once it meets all outstanding technical requirements.

Kind regards,

Muhammad Salman Bashir, M.S.C

Academic Editor

PLOS One

Additional Editor Comments (optional):

Reviewers' comments:

Reviewer's Responses to Questions

**Comments to the Author**

Reviewer #2: All comments have been addressed

Reviewer #3: All comments have been addressed

2. Is the manuscript technically sound, and do the data support the conclusions?

Reviewer #2: Yes

Reviewer #3: Yes

3. Has the statistical analysis been performed appropriately and rigorously?

Reviewer #2: I Don't Know

Reviewer #3: Yes

4. Have the authors made all data underlying the findings in their manuscript fully available?

Reviewer #2: Yes

Reviewer #3: (No Response)

5. Is the manuscript presented in an intelligible fashion and written in standard English?

Reviewer #2: Yes

Reviewer #3: (No Response)

Reviewer #2: This is a Great work , as well as a comprehensive and interesting study about the association between adult onset asthma and BMI, in a large cohort of subjects. Our minör point was the mean BMI of healthy controls was not matched to patient group! The authors replied that they conducted BMI-adjusted models to estimate the direct effects of the BMI-GRS by comparing non-asthmatic healthy controls with patients with asthma. This analysis also confirmed that asthma is not associated with the BMI-GRS, even when adjusted for BMI (p = 0.87), and was mentioned in the result section of the revised manuscript

Reviewer #3: All of the comments and criticisms of the reviewers have been carefully and fully addressed. The study showed that BMI-associated genetic variants showed differential distribution patterns across six distinct asthma clusters, with elevated BMI-GRS observed in both obese and non-obese phenotypes. The manuscript is well written.

**Do you want your identity to be public for this peer review?** For information about this choice, including consent withdrawal, please see our Privacy Policy

Reviewer #2: No

Reviewer #3: No

---

## [Editor Report · Acceptance letter]

PONE-D-25-02383R1

PLOS One

Dear Dr. Yatagai,

I'm pleased to inform you that your manuscript has been deemed suitable for publication in PLOS One. Congratulations! Your manuscript is now being handed over to our production team.

Kind regards,

on behalf of

Dr. Muhammad Salman Bashir

Academic Editor

PLOS One